# Toxic Psychosocial Stress, Resiliency Resources and Time to Dementia Diagnosis in a Nationally Representative Sample of Older Americans in the Health and Retirement Study from 2006–2016

**DOI:** 10.3390/ijerph19042419

**Published:** 2022-02-19

**Authors:** Allan K. Nkwata, Ming Zhang, Xiao Song, Bruno Giordani, Amara E. Ezeamama

**Affiliations:** 1Survey Research Center, Institute of Social Research, University of Michigan, Ann Arbor, MI 48106, USA; nkwata@umich.edu; 2Department of Epidemiology and Biostatistics, University of Georgia, Athens, GA 30602, USA; mzhang01@uga.edu (M.Z.); xsong@uga.edu (X.S.); 3Department of Psychiatry, University of Michigan, Ann Arbor, MI 48105, USA; giordani@med.umich.edu; 4Department of Psychiatry, College of Osteopathic Medicine, Michigan State University, East Lansing, MI 48824, USA

**Keywords:** toxic stress, resilience-promoting factors, incident dementia, everyday discrimination, older Americans

## Abstract

Background: Toxic stress (TS), resiliency-promoting factors (RPFs) and their interactions were investigated in relationship to incident dementia in a nationally representative sample (*n* = 6516) of American adults ≥50 years enrolled in the Health and Retirement Study between 2006 and 2016. Methods: TS included experiences of everyday discrimination and RPF included personal mastery. Race/ethnicity was self-reported as African American, Caucasian, or Other. Multivariable Cox proportional hazards regression models estimated TS-, RPF- and race-associated hazard ratios (HR) for dementia diagnosis and 95% confidence intervals (CIs) with adjustment for comorbidity, lifestyle, and socio-demographic confounders. Results: Discrimination-associated risk of dementia diagnosis on average increased with education level [discrimination x education, *p* = 0.032; HR = 1.75 (95% CI: 1.01–3.03) if < high school, HR = 5.67 (95% CI: 2.94–10.94) if high school completed and HR = 2.48 (95% CI: 1.53–4.00) if ≥some college education]. Likewise, African American vs. Caucasian race disparity in new-onset dementia was evident (HR = 2.12, 95% CI: 1.42–3.17) among adults with high-mastery while absent (HR = 1.35, 95% CI: 0.75–2.41) among adults with low mastery (Mastery x Race, *p* = 0.01). Conclusions: TS is a contextual driver of incident dementia that seemingly operates in a race and RPF-dependent fashion among American adults. Association pattern suggests that TS may overwhelm the cognitive reserve benefit of RPF particularly in status-inconsistent contexts including persons subjected to discrimination despite high education and persons of African American descent despite high mastery. Policies that reduce discrimination and promote equitable treatment by race/ethnicity may support cognitive resiliency and reduce the risk of dementia diagnosis in adult Americans.

## 1. Introduction

Age-associated cognitive impairment—which includes Alzheimer’s dementia, is the sixth leading cause of death globally [1] and in the United States [2] and a growing public health challenge for which effective therapeutic interventions are currently lacking [2]. The global prevalence of dementia is approximately 7% amongst individuals aged 65 and above [3]. In the USA, at least 6 million people are affected by dementia and this number is projected to increase to 14 million people by 2060. Furthermore, in 2014, the prevalence of dementia was higher in African Americans and Hispanics than in non-Hispanic Whites (14.3% and 12.9% vs. 11.3%, respectively) [4]. Premature cognitive decline directly limits workforce participation, the ability to enjoy retirement, diminishes the quality of life, and leads to rapid loss of independence, decreased healthy life expectancy, and elevated mortality [5,6,7,8]. Alzheimer’s dementia leads to institutionalization and increases social and healthcare services utilization at a high cost to direct caregivers and the larger society [9,10,11]. 

The potential contributory role of psychosocial adversity (i.e., toxic stress (TS)), in the incidence of dementia among adults in the United States, remains unknown. Studies of laboratory animals have shown that psychological stress can lead to cellular changes in regions of the hippocampus, decreased proliferation of neurons in the dentate gyrus, and loss of hippocampal volume, resulting in atrophy and cognitive deficits [12,13,14]. In humans, early life stress (e.g., childhood adversity or trauma exposure) has been associated with enduring neuropsychiatric effects, such as depression [15,16] and long-term deficits in cognitive function [17]. Additionally, chronic stress in adults is associated with hormonal and inflammatory indicators of accelerated aging [18], lower quality of life [19], as well as an excess risk of cardiovascular morbidity and mortality, including increased stroke [20,21,22,23]. Equally relevant, a large body of literature suggests that resilience in the face of adversity is associated with well-being [24,25,26], but specific information with respect to age-associated cognitive decline is lacking. Available insight on the potential importance of resilience arises from studies of children who experienced various forms of trauma in early life (i.e., poverty and chronic maltreatment) [27,28]. However, hardly any studies have assessed the joint effects of toxic stress and resilience-promoting factors (RPF) in the development of dementia. 

This research is grounded in the allostatic theory, which theorizes that adversity from chronic stress accelerates both physiological and psychological responses, thus increasing allostatic load, which in turn leads to increased morbidity and mortality from chronic conditions, such as dementia [29]. This study informs an existing knowledge gap regarding potential etiologic roles of TS and RPF in new-onset dementia in an ethnically diverse sample of dementia-free U.S. adults followed every two years with serial cognitive testing and health evaluation in the Health and Retirement Study (HRS). All things being equal, we hypothesized that: (a) higher levels of TS and lower levels of RPF will be associated with an earlier dementia diagnosis in older American adults, (b) race ethnicity will not be associated with earlier onset of cognitive decline over 10 years, and (c) the relationship between race/ethnicity and earlier onset of dementia will vary according to levels of TS and resilience-promoting resources.

## 2. Materials and Methods

### 2.1. Study Population

This was a prospective cohort study from a nationally representative sample of Americans ≥50 years old followed as part of the HRS, along with their spouses/partners who may be younger than 50 years old [30]. Individuals are biennially assessed with detailed assessments of health outcomes, health expenditures, psychosocial and lifestyle factors, employment, retirement, and finances to address issues related to aging Americans. This analysis includes 10 years of data collected as part of six HRS waves spanning 2006 through 2016. 

#### Inclusion and Exclusion Criteria

Inclusion criteria included: (a) availability of data regarding physician diagnosis of dementia and completion of the Psychosocial Leave-Behind (PLB) Participant Lifestyle Questionnaire, which provided data on toxic stress and RPF; (b) absence of physician-diagnosed dementia or memory problems at baseline. Since the PLB questionnaire was administered biennially to all HRS participants in a rotating fashion starting with 50% random being surveyed in 2006 and the remaining 50% not previously surveyed receiving the same questionnaire in 2008 [31], the baseline for individuals in this study was either 2006 or 2008 depending on when the participant completed the PLB questionnaire. 

Participants were excluded for the following reasons: if date of birth unknown, ineligible for the face-to-face interview with the psychosocial leave behind questionnaire, missing more than two stress/resilience measures, or having a diagnosis of a memory-related problem prior to 2006 or 2008. The study base for this study included 6516 respondents as described in Figure 1.

### 2.2. Measures

#### 2.2.1. Primary Determinants: Psychosocial Factors

Psychosocial factors operationally defined in this study as toxic stress (TS) and resilience-promoting factors (RPF) were the risk factors of interest in this study. TS was operationally defined as: cumulative stressors, life course stressors, recent stressors, experiences of everyday discrimination, major experiences of lifetime discrimination, ongoing chronic stressors, and perceived constraints on personal control [31]. Resilience-promoting factors (RPF) on the other hand included global mastery, domain-specific control of finances, health and social life, and measures of positive and negative social support from spouses, children, relatives, friends, and all relationship groups combined. 

Briefly, life course stressors were eleven items that captured stressful life events at any point in a person’s lifetime, such as loss of a child, being in a major fire, flood, earthquake, or other natural disasters, life-threatening illness, or major accident. Recent stressors were six items that captured major stressful life events that occurred in the last five years such as involuntary job loss, prolonged unemployment, being robbed or burglarized, moving to a worse neighborhood, or being a victim of fraud. Cumulative stress is a summation of recent stressors and life-course stressors [19,23,31]. Ongoing chronic stressors were eight items (Cronbach’s alpha = 0.67) that captured current and ongoing problems that have lasted twelve months or longer such as health problems, difficulties at work, housing problems, and financial strain [31,32]. Measures of everyday discrimination were six questions (Cronbach’s alpha = 0.83) that tapped into the hassles and chronic stress associated with perceived everyday discrimination and comprised “character assaults” that tend to occur daily. Major experiences of lifetime discrimination were seven questions that captured major experiences of unfair treatment at any point in one’s lifetime. Experiences of chronic work discrimination (Cronbach’s alpha = 0.85) were designed to assess chronic discrimination experienced at work during the last 12 months [19,31,32]. Perceived constraints were five items (Cronbach’s alpha = 0.87) that captured a sense of lack of control of things going on around an individual [31,32].

Global mastery included five items (Cronbach’s alpha = 0.91) on a person’s resolve at attaining goals. Domain-specific mastery of health, social life, and finances was measured via a single-item measure assessing the amount of control for each aspect on a 10-point scale that ranged from ‘‘no control at all’’ to ‘‘very much control’’, with higher scores indicating greater domain-specific mastery. Measures of social support included four sets of seven items (Cronbach’s alpha = 0.75–0.86) that examined the level of social support received from spouses/partners, children, other family members, and friends. For each relationship category were three positively worded items—positive social support (PSS) and four negatively worded items—negative social support (NSS) [31,32,33]. Details regarding the items for each TS and RPF measure above as well as guidelines on how each is scored have been described elsewhere [19,23,31,32,33]. 

We analyzed each TS and RPF measure first as continuous variables and farther dichotomously based on distributions of events. For instance, for continuous variables, scores ranged from 0–17 for cumulative stressors, 0–11 for life-course stressors, and 0–6 for recent stressors. For categorical variables, cumulative, recent, and life-course stress categories included zero events (reference), and one or more events. This categorization also applied to measures of everyday, chronic work, and lifetime discrimination. However, ongoing chronic stressors, personal constraints, and RPF measures were dichotomized as high vs. low based on their mean distributions [19,31,32,33]. 

#### 2.2.2. Other Measures

Additional factors were measured at baseline and were included in the analyses based on a review of the literature and what HRS collects. Socio-demographic factors included race, sex, retirement status, education level, and marital status. Lifestyle covariates included alcohol use, tobacco use, and moderate physical activity. BMI and comorbidities were also assessed. Comorbidities included the following physician-diagnosed conditions: high blood pressure, diabetes, stroke, lung problems (i.e., chronic bronchitis or emphysema), arthritis, psychiatric problems (i.e., emotional, or nervous), and cancer. A comorbidity index was created where one point was given for each yes with a maximum total of seven. 

#### 2.2.3. Outcome: Assessment of Dementia

The main outcome was incident dementia defined by a new dementia diagnosis and time to that new diagnosis. A new dementia diagnosis was defined by a ‘no’ diagnosis in 2006 and a report of ‘yes’ in any of the subsequent years since the last interview based on the question, “Has a doctor ever told you that you have a memory-related disease?” In 2010, this question was changed to ask participants if they have ever been told they have Alzheimer’s disease or dementia. Time to new dementia was defined as participant chronologic age in the first year a new dementia diagnosis was reported. Participants with a ‘no’ in all interviews were censored in the 2016 study year. 

### 2.3. Statistical Analyses

Race, TS, and RPF were analyzed as predictors of new-onset dementia over ten years of follow-up. First, descriptive analyses determined the distribution of baseline TS, RPF, and race/ethnicity. Bivariate associations were implemented to determine crude associations for TS, RPF, potential confounders, and sociodemographic factors with race. Since both TS and RPF were analyzed as categorical variables, chi-square tests were used to evaluate differences in proportions by race/ethnicity. Kaplan–Meier curves for differences in dementia-free survival times were generated according to categories of TS and RPF measures and compared using the log-rank test. Factors with a *p*-value ≤ 0.2 were further evaluated in multivariable models as candidate confounders. 

Cox proportional hazards regression models were used to evaluate the association between TS/RPF parameters and incident dementia. Hazard ratios (HR) were generated and reported with 95% confidence intervals (CI). Proportional hazards assumptions were assessed by graphing log–log survival curves and examining Schoenfeld residuals. The following covariates were assessed as candidate confounders: race/ethnicity, sex, education status, alcohol consumption, moderate physical activity, BMI, retirement status and comorbidity due to diabetes, heart diseases, and stroke. A series of incremental nested models were implemented, beginning with crude models, followed by models with sociodemographic factors adjusted for, and models adjusting for sociodemographic factors as well as TS. The final models were further adjusted for RPF. Measures of TS and RPF were not mutually adjusted for one another in any multivariable model. Additionally, separate Cox regression models evaluated the interaction between race/ethnicity and respective TS and RPF and the potential for interaction between TS and RPF. *p*-values for interaction effects were set at *p* < 0.10 because the power of statistical tests for higher-order terms is generally lower than for first-order terms [34,35]. All results were adjusted for the complex sampling design of the HRS [36]. All analyses were implemented with SAS software, version 9.4 (SAS Institute, Cary, NC, USA). 

## 3. Results

A total of 6516 unique individuals were included in this analysis. The sample included 83% non-Hispanic White, 13% Black/African American, and 4% Other race (including Hispanics). The majority of the sample were female (63%), 70% were married/partnered, 45% had some college education or more, 33% had a cardiometabolic diagnosis (HD, T2DM, or stroke). Of note, 27% reported ≥ three comorbid conditions. Over the ten-year follow-up period, 338 (5%) individuals reported being diagnosed with dementia. 

The proportions of individuals that reported experience of everyday discrimination (*p* < 0.0001), major experiences of lifetime discrimination (*p* < 0.0001), chronic stressors (*p* = 0.0039), and perceived constraints (*p* = 0.0016) were higher amongst individuals that identified as minority relative to non-Hispanic White ethnicity (Table 1). Additional baseline data on the distribution of sociodemographic and lifestyle factors by race are reported in Appendix A. 

After adjusting for race, sex, retirement status, education, marital status, moderate physical activity, smoking, alcohol consumption, BMI, and comorbidity due to heart disease, type 2 diabetes, and stroke, high levels of ongoing chronic stress were associated with shorter time to onset of new dementia (76 vs. 80 years) with 99 times higher risk (HR 1.99, 95% CI: 1.37–2.88) of developing dementia in comparison to individuals with low chronic stress. Similarly, the risk of developing dementia was 60% higher (HR 1.60, 95% CI: 1.02–2.52) amongst individuals that reported increased chronic stress relative to baseline. Higher levels of negative social support (NSS) from all relationship groups were associated with a 68% increased risk of new-onset dementia (HR 1.68, 95% CI: 1.22–2.33). The risk of developing dementia was 48% higher among older Americans who reported an increase in NSS from all relationship groups relative to baseline (HR 1.48, 95% CI: 1.03–2.11, Table 2). Additionally, a trend of shorter dementia diagnosis-free survival time was evident in those that reported one or more experiences of everyday discrimination (*p* < 0.0001, Figure 2a), and one or more major experiences of lifetime discrimination (*p* = 0.0035, Figure 2b). 

### 3.1. Race Is Associated with Incident Dementia, and Disparities Persist According to Level of Mastery

Unadjusted for confounders, the risk of incident dementia was 86% higher in African Americans (HR 1.86, 95% CI: 1.39–2.48) and 79% higher in Other race (HR 1.79, 95% CI: 0.96–3.37) compared to Caucasians over 10 years. This association was down modulated in multivariable models, with the risk of developing dementia attenuated though remained statistically robust (HR 1.70, 95% CI: 1.20–2.40) among African Americans relative to Caucasians (Table 2). We also found that African Americans advanced to dementia on average a year earlier than their Caucasian colleagues (78.8 vs. 79.7 years, respectively: data not shown). However, race-related differences in risk of incident dementia were dependent on the level of mastery amongst these older Americans (mastery x race, *p* = 0.01, Figure 3). Among African Americans, having high mastery was not associated with the onset of dementia. However, among Caucasians, high mastery was associated with 21% protection from the risk of dementia (HR 0.79, 95% CI:0.61, 1.02). Among those with low mastery, there was no association between African Americans and Caucasians in risk of dementia. However, among those with high mastery, the risk of incident dementia was 112% higher in African Americans relative to Caucasians (HR 2.12, 95% CI: 1.42–3.17, Figure 3). Furthermore, among African Americans, those with low mastery on average developed dementia 5 years faster than those with high mastery (76.4 vs. 81.4 years, respectively).

### 3.2. Toxic Stress Is Associated with Incident Dementia; Relationship Varies by Level of Mastery

The relative hazard of new-onset dementia diagnosis over 10 years follow-up from unadjusted models was 43% elevated (HR 1.43, 95% CI: 1.04–1.98) for older adults reporting ≥1 vs. no cumulative stress events. The strength of this association weakened with adjustment for sociodemographic factors, lifestyle factors, and comorbidity (Table 2), but varied according to levels of mastery (cumulative stress x mastery, *p* = 0.082). Among older adults for whom cumulative stress events were zero, i.e., no stress was reported; there was no association of cumulative stress to incident dementia between those with high vs. low mastery. However, among those that reported ≥1 cumulative stress events, having high mastery was associated with 29% protection from the risk of dementia (HR 0.71, 95% CI: 0.53–0.96). 

Conversely, among older adults categorized as having low mastery level, the relative hazard of dementia diagnosis was 87% elevated for adults reporting ≥1 vs. zero cumulative stress events (HR 1.87, 95% CI: 1.16–3.00). This relative hazard among adults in the low mastery category corresponds to an average of 2.5 years earlier dementia onset for individuals reporting ≥1 vs. zero cumulative stress (- i.e., 79.5 vs. 82.0 years, respectively). Among older adults classified as having high mastery, the experience of ≥1 vs. zero cumulative stress events was not associated with incident dementia diagnosis (HR 1.04, 95% CI: 0.66–1.65, Figure 4). This relative hazard on average corresponded to 1.9 years earlier dementia diagnosis for high mastery adults reporting ≥1 vs. zero cumulative stress events (79.7 vs. 81.6 years, respectively). 

### 3.3. Toxic Stress Is Associated with Incident Dementia: Relationship Varies by Level of Education

Experiences of everyday discrimination were strongly associated with incident dementia in unadjusted models (HR 2.84, 95% CI: 2.03–3.96). This association remained elevated with adjustment for confounding variables for older Americans reporting any vs. no experiences of everyday discrimination (HR 2.74, 95% CI: 1.89–3.98). Additionally, the risk of developing dementia was 165% higher amongst individuals that experienced increased discrimination above their baseline (HR 2.65, 95% CI: 1.32, 5.29, Table 2). Similarly, having a lower education was associated with incident dementia in unadjusted models (HR 1.81, 95% CI: 1.34–2.44). This association persisted after adjusting for confounding variables for older adults with less than high school having a 52% increased risk for incident dementia (HR 1.52, 95% CI: 1.09–2.11, Appendix A). 

However, the relationship of everyday discrimination to incident dementia varied according to education status (discrimination x education, *p* = 0.032). Among older adults that did not experience everyday discrimination, the risk of new-onset dementia diagnosis over 10 years was 1.88 times higher for those with less than high school education (HR 1.88, 95% CI: 1.26–2.81) relative to those that were high school graduates, or 1.59 times elevated for adults with less than high school (HR 1.59, 95% CI: 1.12–2.26) vs. peers with college or higher levels of education. Furthermore, among older adults that reported no experience of everyday discrimination, individuals with less than high school education advanced to dementia onset 2 years on average relative to those with higher-level education (78.6 vs. 80.6 years, respectively). 

Among older adults that reported ≥1 experience of everyday discrimination, on the other hand, education was not associated with dementia incidence over 10 years (Figure 5). Amongst individuals in this group, those with less than high school advanced to dementia onset 2–6 years on average relative to those with higher-level education (78.3 vs. 80.1 years-high school, 84.3 years-college and above). 

Among older adults with less than high school, experiencing any vs. no discrimination was associated with a 75% increased risk of dementia, (HR 1.75, 95% CI: 1.01–3.84). Among high school graduates, the risk of incident dementia was amplified 467% higher for older adults reporting any (HR 5.67, 95% CI: 2.94–10.94) vs. no experiences of everyday discrimination. Similarly, among older adults with college or higher-level education, any vs. no experience of everyday discrimination was associated with a 148% higher incidence of dementia (HR 2.48, 95% CI: 1.53–4.00, Figure 5). 

### 3.4. Other Factors Associated with Incident Dementia 

Cigarette smoking, alcohol consumption, and retirement status, as well as comorbid stroke diagnosis, were each associated with earlier age at dementia onset, independent of TS, RPF, sociodemographic, and lifestyle factors (Appendix A).

## 4. Discussion

In this cohort of aging American adults that were dementia-free at enrollment and followed for ten years through 2016, we found that higher levels of toxic stressors, including everyday discrimination, ongoing chronic stressors and perceived constraints at baseline were each associated with younger age at dementia diagnosis. These associations were independent of several sociodemographic confounders, such as sex, marital status, retirement status, BMI, comorbidity, smoking status, and alcohol consumption. Furthermore, we found novel empirical evidence that in the presence of discrimination, the benefit of education for cognitive reserve is muted. These findings are in line with our study hypothesis that TS in aging adults will be associated with an earlier dementia diagnosis. They also corroborate findings from other studies that linked stressful conditions, socioeconomic and social disadvantage, whether defined as low education, limited income, living in a disadvantaged neighborhood or exposure to racial discrimination, to accelerated aging [37], and to early onset of illness and death [38]. They also align with our cross-sectional observation that higher levels of toxic stressors and lower levels of resilience resources were associated with an increased risk for neurocognitive impairment among older adults in the HRS [32]. 

Several studies have reported that individuals of African American race are at higher risk of dementia [39,40,41,42]. By observing the persistence of racial disparity with African Americans at higher risk of new-onset dementia among individuals with high mastery, this data provides partial support for these prior reports. Data from this study further suggest that while the overall risk in dementia onset by race/ethnicity was limited, racial differences persisted within levels of mastery for African Americans. Our finding of no overall risk of dementia by race/ethnicity is consistent with some, but not all previous literature on differences in cognitive decline by race [43,44,45]. However, the finding on racial differences within levels of mastery highlights the disparities in social experiences, such as racism in all its forms that exacerbate cognitive function in Older African Americans vs. Caucasians, as African American vs. White race-associated disadvantage in dementia incidence was evident amongst Older Americans with high mastery. This attenuates the would-be beneficial effects of high mastery for African Americans and is similar to what we reported in our earlier study [32]. Furthermore, our finding associating earlier advancement to dementia in African Americans compared to Caucasians has been corroborated in some, but not all studies that evaluated racial/ethnic differences in cognitive function [46,47,48]. 

Findings from our study also showed that psychosocial adversity-associated risk for dementia onset in these older adults varied according to levels of mastery. This shows that mastery is protective in the face of adversity, and that mastery is associated with cognitive reserve. This is consistent with a study that found that individuals that had high levels of resilience traits showed less distress despite reported childhood adversities relative to those that had low resilient coping abilities [49].

Data from this study further suggest the deleterious effects of everyday discrimination on education. Education has been associated with several beneficial effects that include building cognitive reserve-enhanced reasoning skills, test-taking abilities, verbal and working memory—all of which translate to personal mastery [50,51], better health behaviors, income, and social opportunities [52]. Our findings, however, suggest that the systemic structures that perpetuate racism and discrimination overwhelm the benefits of education for African Americans, thus adversely affecting health outcomes [53,54]. This confirms prior research where we found that individuals who experienced discrimination regardless of mastery, had an elevated risk of neurocognitive impairment [32]. Additionally, another study reported higher levels of allostasis for Black and Mexican Americans relative to White Americans with a college degree or higher, whereas allostasis was similar across race groups among adults with low educational achievement in the same study [55]. Prior data shows that Black Americans of higher educational status report a high frequency of experienced micro-aggression and workplace discrimination and more frequently report being in jobs below their qualification level [56]. Both the nature and frequency of everyday discrimination varies according to race, with African Americans more frequently on the receiving end of the most insidious forms of discrimination in occupational and social interactions—whether in healthcare, educational, financial, law enforcement by police, and judicial systems [56,57].

This study raises awareness of the influence that social determinants of health have on the development of cognitive impairment especially in African American communities. The World Health Organization’s Commission on Social Determinants of Health states that the high burden of illness leading to premature death is a result of the conditions in which people are born, grow, age, work, and live [58]. It is important to address the conditions that shape a person’s well-being during all stages of life. A person’s well-being is multidimensional and involves dimensions such as health, education, environment, work, and physical insecurity. The domains of stressful events evaluated in this study involve these areas of a person’s well-being and therefore play a role in how a person responds to the development of conditions like dementia. 

Some of the strengths in this study include the implementation of a large nationally representative prospective, study design using rigorous analytic approaches adjusted for multiple confounders. Additionally, we evaluated multiple indicators of TS and RPF as proxies for social experiences that may affect cognitive aging. However, there are limitations to consider when interpreting our results. Self-reported assessments of psychosocial factors and dementia diagnosis were used, allowing for potential information bias and recall bias despite meticulous efforts made to collect data in a standardized method. HRS data were collected biennially, and the assessment of dementia was within the previous two years. Potential misclassification could have occurred with the time of dementia diagnosis, but it should not affect the association between baseline psychosocial factors and dementia over ten years. Lastly, dementia outcome was based on a general definition that changed in 2010 to include dementia and Alzheimer’s disease and did not allow for assessing risk for specific Alzheimer’s-related conditions. 

## 5. Conclusions

This study provided further empirical evidence that high psychosocial adversity and low levels of RPF are important social determinants of cognitive impairment in a diverse sample of older US adults. Psychosocial processes grounded in structural inequity sustain pockets of racial disparities in cognitive function in older Americans. African American race was associated with cognitive disadvantage, but only in the status inconsistent context of high mastery. Regardless of race, the benefit of high mastery for cognitive reserve among older Americans was muted among those that reported experience of discrimination. Similarly, regardless of race, the benefit of education for cognitive reserve was virtually absent among older Americans that reported experiencing discrimination. This pattern of heterogeneity suggests that policies/structural interventions that reduce discrimination and promote equitable treatment by race/ethnicity in addition to reducing toxic psychosocial stress may delay time to onset of dementia by allowing a broader section of American adults to reap the cognitive-reserve benefit of higher mastery and educational attainment. 

## Figures and Tables

**Figure 1 ijerph-19-02419-f001:**
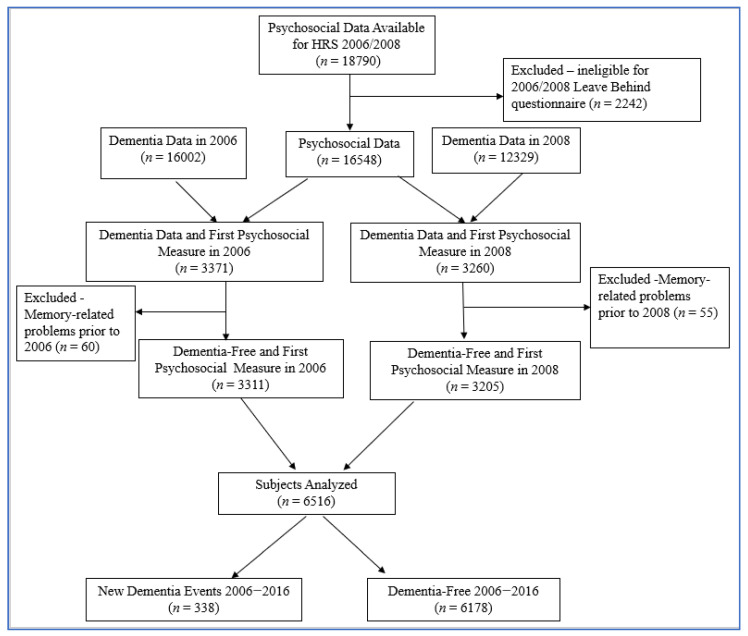
Sample selection to assess the association between Dementia and Psychosocial measures in the Health and Retirement Study, 2006–2016.

**Figure 2 ijerph-19-02419-f002:**
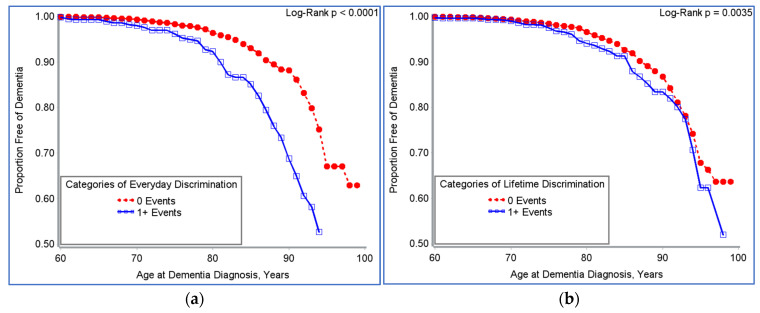
(**a**) Dementia-free survival time by experiences of everyday discrimination among participants in the HRS, 2006–2016; (**b**) dementia-free survival time by major experiences of lifetime discrimination among participants in the HRS, 2006–2016.

**Figure 3 ijerph-19-02419-f003:**
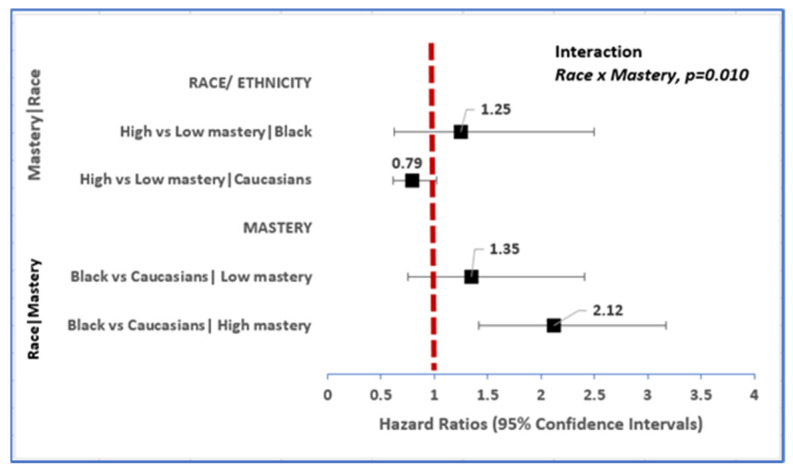
Race-related differences in risk of incident dementia vary within strata of mastery.

**Figure 4 ijerph-19-02419-f004:**
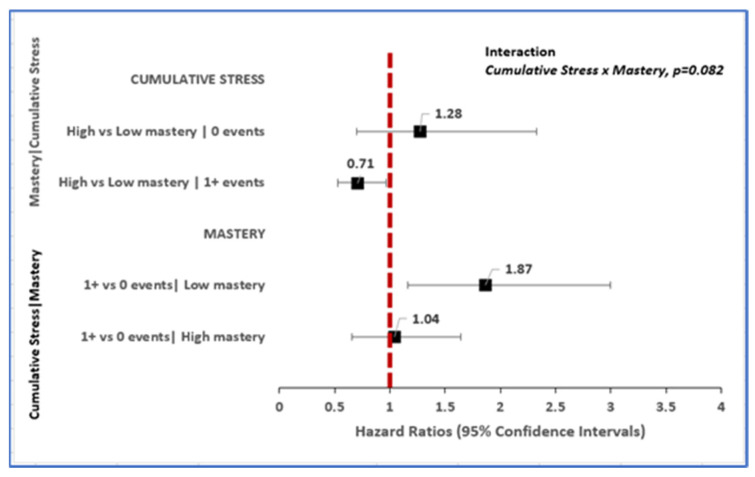
Cumulative stress-related differences in risk of incident dementia vary within strata of mastery.

**Figure 5 ijerph-19-02419-f005:**
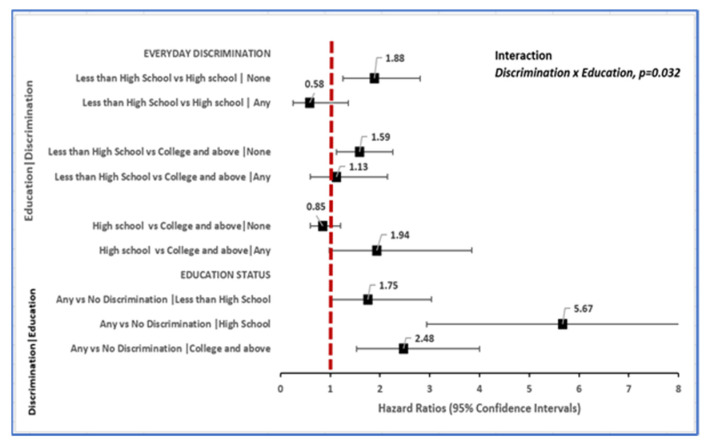
Discrimination-related differences in risk of incident dementia within strata of education.

**Table 1 ijerph-19-02419-t001:** Distribution of toxic stress and resilience promoting factors among older Americans enrolled in the HRS 2006–2016 sample at baseline by race/ethnicity.

Characteristic	All (N = 6516)	White/Caucasian (N = 5440)	Black/African American (N = 844)	Other (N = 232)	
Dimensions of Toxic Stress	N (%)	N (%)	N (%)	N (%)	*p*-Value
Cumulative stress ^a^					
Median (IQR)	2.0 (1.0, 3.0)	2.0 (1.0, 3.0)	1.0 (1.0, 3.0)	1.0 (1.0, 3.0)	
0 events	1358 (22.0)	1143 (21.9)	167 (22.2)	48 (23.1)	0.9104
1+ events	4817 (78.0)	4073 (78.1)	584 (77.8)	160 (76.9)	
Life course stress					
Median (IQR)	1.0 (0.0, 2.0)	1.0 (1.0, 2.0)	1.0 (0.0, 2.0)	1.0 (0.0, 2.0)	
0 events	1547 (25.0)	1299 (24.8)	193 (25.6)	55 (26.4)	0.7962
1+ events	4642 (75.0)	3929 (75.2)	560 (74.4)	153 (73.6)	
Recent Stress					
Median (IQR)	0.0 (0.0, 0.0)	0.0 (0.0, 0.0)	0.0 (0.0, 0.0)	0.0 (0.0, 0.0)	
0 events	5084 (81.8)	4320 (82.3)	598 (78.8)	166 (78.7)	0.0292
1+ events	1132 (18.2)	926 (17.7)	161 (21.2)	45 (21.3)	
Everyday Discrimination					
Median (IQR)	1.0 (0.0, 1.0)	0.0 (0.0, 0.0)	0.0 (0.0, 0.0)	0.0 (0.0, 0.0)	
0 events	5722 (90.7)	4888 (92.2)	649 (82.0)	185 (85.3)	<0.0001
1+ events	590 (9.3)	416 (7.8)	142 (18.0)	32 (14.7)	
Lifetime Discrimination					
Median (IQR)	0.0 (0.0, 1.0)	0.0 (0.0, 1.0)	0.0 (0.0, 1.0)	0.0 (0.0, 1.0)	
0 events	4397 (70.8)	3802 (72.6)	445 (58.8)	150 (71.1)	<0.0001
1+ events	1810 (29.2)	1437 (27.4)	312 (41.2)	61 (28.9)	
Chronic work-Discrimination ^b^					
Median (IQR)	0.0 (0.0, 0.0)	0.0 (0.0, 0.0)	0.0 (0.0, 0.0)	0.0 (0.0, 0.0)	
0 events	2852 (91.5)	2390 (92.3)	356 (87.7)	106 (85.5)	0.0004
1+ events	266 (8.5)	198 (7.7)	50 (12.3)	18 (14.5)	
Chronic stress					
Median (IQR)	3.0 (3.0, 4.0)	3.0 (3.0, 4.0)	3.0 (3.0, 4.0)	3.0 (2.0, 4.0)	
Low	4122 (70.6)	3519 (71.5)	473 (66.6)	130 (64.4)	0.0039
High	1712 (29.4)	1403 (28.5)	237 (33.4)	72 (35.6)	
Perceived constraints					
Median (IQR)	2.0 (1.0, 3.0)	2.0 (1.0, 3.0)	2.0 (1.0, 3.0)	2.0 (1.0, 4.0)	
Low	4214 (66.9)	3561 (67.3)	534 (67.5)	119 (55.6)	0.0016
High	2081 (33.1)	1729 (32.7)	257 (32.5)	95 (44.4)	
Measures of resilience					
Personal mastery					
Median (IQR)	5.0 (4.0, 6.0)	5.0 (4.0, 6.0)	5.0 (4.0, 6.0)	5.0 (4.0, 6.0)	
Low	2189 (34.7)	1850 (34.9)	273 (34.5)	66 (30.7)	0.4374
High	4113 (65.3)	3446 (65.1)	518 (65.5)	149 (69.3)	
Control over health					
Median (IQR)	8.0 (6.0, 9.0)	8.0 (6.0 9.0)	8.0 (6.0, 9.0)	8.0 (7.0, 9.0)	
Low	2795 (44.8)	2381 (45.4)	336 (43.1)	78 (36.8)	0.0281
High	3444 (55.2)	2866 (54.6)	444 (56.9)	134 (63.2)	
Control over finances					
Median (IQR)	8.0 (6.0, 10.0)	8.0 (6.0, 10.0)	8.0 (6.0, 10.0)	9.0 (7.0, 10.0)	
Low	2207 (35.2)	1878 (35.6)	263 (33.4)	66 (31.0)	0.2085
High	4064 (64.8)	3393 (64.4)	524 (66.6)	147 (69.0)	
Control over social life					
Median (IQR)	8.0 (7.0, 10.0)	8.0 (7.0, 10.0)	9.0 (7.0, 10.0)	9.0 (7.0, 10.0)	
Low	1899 (30.4)	1630 (31.0)	207 (26.9)	62 (29.2)	0.0637
High	4342 (69.6)	3629 (69.0)	563 (73.1)	150 (70.8)	
Positive Social Support (PSS)					
PSS: spouses/partners					
Median (IQR)	4.0 (3.0, 4.0)	4.0 (3.0, 4.0)	3.0 (3.0, 4.0)	3.0 (3.0, 4.0)	
Low	1882 (42.1)	1569 (40.6)	230 (51.8)	83 (51.2)	<0.0001
High	2587 (57.9)	2294 (59.4)	214 (48.2)	79 (48.8)	
PSS: children					
Median (IQR)	3.0 (3.0, 4.0)	3.0 (3.0, 4.0)	4.0 (3.0, 4.0)	3.5 (3.0, 4.0)	
Low	2979 (50.6)	2559 (51.7)	320 (43.1)	100 (50.3)	<0.0001
High	2910 (49.4)	2388 (48.3)	423 (56.9)	99 (49.7)	
PSS: Other family members					
Median (IQR)	3.0 (2.0, 4.0)	3.0 (2.0, 4.0)	3.0 (3.0, 4.0)	3.0 (2.0, 4.0)	
Low	4268 (70.0)	3666 (71.8)	454 (58.3)	148 (70.5)	<0.0001
High	1827 (30.0)	1441 (28.2)	324 (41.7)	62 (29.5)	
PSS: Friends					
Median (IQR)	3.0 (3.0, 4.0)	3.0 (3.0, 4.0)	3.0 (3.0, 4.0)	3.0 (2.0, 4.0)	
Low	4172 (68.8)	3539 (69.3)	484 (64.6)	149 (71.6)	0.0391
High	1889 (31.2)	1565 (30.7)	265 (35.4)	59 (28.4)	
PSS: all relationship groups					
Median (IQR)	8.0 (7.0, 10.0)	8.0 (7.0, 10.0)	8.0 (7.0, 9.0)	8.0 (7.0, 9.0)	
Low	3502 (55.3)	2897 (54.4)	474 (59.6)	131 (60.4)	0.0076
High	2835 (44.7)	2427 (45.6)	322 (40.4)	86 (39.6)	
Negative Social Support (NSS)					
NSS: spouses/partners					
Median (IQR)	2.0 (2.0, 2.0)	2.0 (2.0, 2.0)	2.0 (2.0, 3.0)	2.0 (1.0, 3.0)	
Low	3377 (75.7)	2948 (76.4)	319 (72.3)	110 (68.7)	0.0196
High	1084 (24.3)	912 (23.6)	122 (27.7)	50 (31.3)	
NSS: children					
Median (IQR)	2.0 (1.0, 2.0)	2.0 (1.0, 2.0)	2.0 (1.0, 2.0)	2.0 (1.0, 2.0)	
Low	2503 (42.3)	2162 (43.5)	260 (34.8)	81 (40.1)	<0.0001
High	3416 (57.7)	2808 (56.5)	487 (65.2)	121 (59.9)	
NSS: other family members					
Median (IQR)	1.0 (1.0, 2.0)	1.0 (1.0, 2.0)	1.0 (1.0, 2.0)	2.0 (1.0, 2.0)	
Low	3459 (56.8)	3015 (59.1)	346 (44.5)	98 (46.9)	<0.0001
High	2631 (43.2)	2088 (40.9)	432 (55.5)	111 (53.1)	
NSS: friends					
Median (IQR)	1.0 (1.0, 2.0)	1.0 (1.0, 2.0)	1.0 (1.0, 2.0)	1.0 (1.0, 2.0)	
Low	3900 (64.4)	3369 (66.1)	409 (54.5)	122 (58.6)	<0.0001
High	2158 (35.6)	1731 (33.9)	341 (45.5)	86 (41.4)	
NSS: all relationship groups					
Median (IQR)	5.0 (4.0, 7.0)	5.0 (4.0, 6.0)	5.0 (4.0, 7.0)	5.0 (4.0, 7.0)	
Low	3612 (57.0)	3065 (57.6)	436 (54.8)	111 (51.1)	0.0711
High	2727 (43.0)	2261 (42.4)	360 (45.2)	106 (48.9)	
Onset of Dementia					0.0959
No	6178 (94.8)	5172 (95.1)	788 (93.4)	218 (94.0)	
Yes	338 (5.2)	268 (4.9)	56 (6.6)	14 (6.0)	

Note: ^a^ Cumulative stress is the sum of recent and life course events, sum can be 0–17. ^b^ Questions on chronic work discrimination asked of only working participants.

**Table 2 ijerph-19-02419-t002:** Race, toxic stress, and resilience promoting factors in relation to risk for incident dementia among older adults enrolled in the HRS 2006–2016.

Characteristic	*n*/N	Unadjusted HR (95% CI)	*p*-Value	Adjusted HR ^c^ (95% CI)	*p*-Value
**Race**					
**Black (AA) vs. Caucasian**	56/844	1.86 (1.39, 2.48)	<0.0001	1.70 (1.20, 2.40)	0.0033
**Other vs. Caucasian**	14/232	1.79 (0.96, 3.37)	0.0674	1.41 (0.61, 3.24)	0.4155
**Toxic stress (TS) measures**					
**Cumulative stress ^d^**					
Continuous measure	298/6106	1.17 (1.09, 1.30)	0.0003	1.14 (1.04, 1.24)	0.0071
Baseline cumulative stress					
0 events	55/1358	1.00		1.00	
1+ events	243/4817	1.43 (1.04, 1.98)	0.029	1.37 (0.97, 1.94)	0.0739
Change in stress (increase vs. no change)	52/1292	1.18 (0.85, 1.63)	0.3192	1.25 (0.88, 1.78)	0.2137
**Life course stress**					
Continuous measure	299/6120	1.18 (1.07, 1.29)	0.0011	1.13 (1.03, 1.24)	0.0133
Baseline life course stress					
0 events	58/1547	1.00		1.00	
1+ events	241/4401	1.47 (1.10, 1.96)	0.0097	1.42 (1.05, 1.92)	0.0244
Change in stress (increase vs. no change)	51/1143	1. 30 (0.92, 1.84)	0.1324	1.40 (0.97, 2.02)	0.0744
**Recent stress**					
Continuous measure	301/6147	1.28 (1.01, 1.64)	0.0427	1.25 (0.97, 1.62)	0.0891
Baseline recent stress					
0 events	254/5084	1.00		1.00	
1+ events	47/1132	1.49 (1.03, 2.16)	0.0389	1.39 (0.95, 2.03)	0.0854
Change in stress (increase vs. no change)	19/613	1.01 (0.60, 1.70)	0.9718	1.01 (0.56, 1.80)	0.9842
**Everyday discrimination**					
Continuous measure	310/6241	1.66 (1.45, 1.89)	<0.0001	1.60 (1.37, 1.87)	<0.0001
Baseline everyday discrimination					
0 events	242/5722	1.00		1.00	
1+ events	68/590	2.84 (2.03, 3.96)	<0.0001	2.74 (1.89, 3.98)	<0.0001
Change in stress (increase vs. no change)	10/148	2.23 (1.11, 4.49)	0.0257	2.65 (1.32, 5.29)	0.0069
**Lifetime discrimination**					
Continuous measure	300/6138	1.23 (1.04, 1.44)	0.0134	1.20 (1.00, 1.44)	0.0556
Baseline lifetime discrimination					
0 events	209/4397	1.00		1.00	
1+ events	91/1810	1.40 (1.07, 1.83)	0.0161	1.33 (0.98, 1.81)	0.0695
Change in stress (increase vs. no change)	17/475	1.49 (0.87, 2.55)	0.1425	1.56 (0.86, 2.85)	0.1399
**Chronic work discrimination**					
Continuous measure	95/3060	1.13 (0.76, 1.68)	0.5388	0.81 (0.41, 1.60)	0.542
Baseline work discrimination					
0 events	84/2852	1.00		1.00	
1+ events	11/266	1.42 (0.73, 2.78)	0.2969	0.92 (0.34, 2.52)	0.8704
**Perceived constraints**					
Continuous measure	310/6224	1.24 (1.13, 1.36)	<0.0001	1.22 (1.10, 1.35)	0.0002
Baseline perceived constraints					
Low constraints	151/4214	1.00		1.00	
High constraints	159/2081	1.61 (1.21, 2.13)	0.0016	1.61 (1.19, 2.18)	0.0024
Change in stress (increase vs. no change)	47/1084	1.45 (0.98, 2.14)	0.0633	1.71 (1.13, 2.58)	0.0115
**Ongoing chronic stressors**					
Continuous measure	227/5765	1.35 (1.16, 1.58)	0.0002	1.31 (1.11, 1.56)	0.0021
Baseline chronic stress					
Low chronic stress	140/4122	1.00		1.00	
High chronic stress	87/1712	2.04 (1.50, 2.79)	<0.0001	1.99 (1.37, 2.88)	0.0005
Change in stress (increase vs. no change)	32/859	1.34 (0.86, 2.10)	0.1911	1.60 (1.02, 2.52)	0.0428
**Resilience-promoting factors (RPF)**					
**Personal mastery**					
Continuous measure	310/6231	0.87 (0.78, 0.97)	0.0117	0.86 (0.77, 0.97)	0.0119
Baseline personal mastery					
Low mastery	152/2189	1.00		1.00	
High mastery	158/4113	0.80 (0.63, 1.02)	0.0723	0.78 (0.59, 1.02)	0.0659
Change in measure (decrease vs. no change)	45/1064	1.12 (0.79, 1.56)	0.533	1.13 (0.81, 1.57)	0.3464
**Positive social support (PSS) domains**					
**PSS from spouses/partners**					
Continuous measure	211/4403	0.92 (0.71, 1.19)	0.5164	0.98 (0.76, 1.28)	0.9069
Baseline PSS (spouse/partner)					
Low PSS	94/1882	1.00		1.00	
High PSS	117/2587	0.94 (0.67, 1.31)	0.7023	1.05 (0.73, 1.50)	0.791
Change in measure (decrease vs. no change)	15/394	1.08 (0.60, 1.93)	0.7954	1.11 (0.60, 2.05)	0.7408
**PSS from children**					
Continuous measure	291/5820	0.77 (0.62, 0.94)	0.0134	0.78 (0.63, 0.97)	0.0276
Baseline PSS (Children)					
Low PSS	138/2979	1.00		1.00	
High PSS	153/2910	0.76 (0.60, 0.95)	0.0182	0.76 (0.61, 0.96)	0.0193
Change in measure (decrease vs. no change)	34/804	1.13 (0.74, 1.71)	0.565	1.26 (0.84, 1.88)	0.2559
**PSS from other family members**					
Continuous measure	295/6024	1.03 (0.90, 1.19)	0.6656	1.00 (0.86, 1.17)	0.9528
Baseline PSS (family)					
Low PSS	201/4268	1.00		1.00	
High PSS	94/1827	0.89 (0.66, 1.18)	0.398	0.82 (0.60, 1.13)	0.2219
Change in measure (decrease vs. no change)	38/947	1.13 (0.73, 1.78)	0.5732	1.24 (0.76, 2.02)	0.3851
**PSS from friends**					
Continuous measure	292/5994	0.99 (0.82, 1.20)	0.9427	1.04 (0.85, 1.27)	0.6909
Baseline PSS (friends)					
Low PSS	206/4172	1.00		1.00	
High PSS	86/1889	0.99 (0.73, 1.33)	0.989	1.04 (0.75, 1.45)	0.8085
Change in measure (decrease vs. no change)	28/824	0.95 (0.56, 1.63)	0.8593	0.99 (0.56, 1.73)	0.9611
**PSS from all relationship groups**					
Continuous measure	314/6265	1.05 (0.99, 1.11)	0.1288	1.03 (0.97, 1.09)	0.3369
Baseline PSS (all groups)					
Low PSS	172/3502	1.00		1.00	
High PSS	142/2835	1.21 (0.93, 1.55)	0.1456	1.04 (0.78, 1.39)	0.7726
Change in measure (decrease vs. no change)	57/1311	1.40 (0.95, 2.06)	0.0902	1.40 (0.92, 2.13)	0.1106
**Negative social support (NSS) domains**					
**NSS from spouses/partners**					
Continuous measure	209/4395	1.13 (0.91, 1.40)	0.2722	1.07 (0.84, 1.36)	0.5814
Baseline NSS (spouse/partner)					
Low NSS	156/3377	1.00		1.00	
High NSS	53/1084	1.17 (0.83, 1.65)	0.3627	1.12 (0.72, 1.74)	0.6077
Change in measure (increase vs. no change)	23/528	1.72 (0.95, 3.12)	0.0731	1.83 (0.91, 3.68)	0.087
**NSS from children**					
Continuous measure	293/5850	1.41 (1.16, 1.72)	0.001	1.35 (1.08, 1.69)	0.0102
Baseline NSS (children)					
Low NSS	102/2503	1.00		1.00	
High NSS	191/3416	1.77 (1.27, 2.46)	0.0011	1.68 (1.16, 2.45)	0.0069
Change in measure (increase vs. no change)	25/772	1.06 (0.65, 1.73)	0.8272	1.12 (0.66, 1.90)	0.6626
**NSS from other family members**					
Continuous measure	293/6019	1.52 (1.24, 1.86)	0.0001	1.43 (1.14, 1.80)	0.0028
Baseline NSS (family)					
Low NSS	156/3459	1.00		1.00	
High NSS	137/2361	1.82 (1.37, 2.42)	<0.0001	1.77 (1.29, 2.44)	0.0007
Change in measure (increase vs. no change)	42/876	1.40 (0.93, 2.10)	0.1038	1.50 (0.99, 2.26)	0.0563
**NSS from friends**					
Continuous measure	292/5991	1.43 (1.12, 1.82)	0.0051	1.33 (1.03, 1.70)	0.0268
Baseline NSS (friends)					
Low NSS	171/3900	1.00		1.00	
High NSS	121/2158	1.54 (1.13, 2.10)	0.0074	1.58 (1.14, 2.21)	0.0073
Change in measure (increase vs. no change)	34/693	1.70 (1.00, 2.89)	0.0495	1.85 (1.12, 3.05)	0.0171
**NSS from all relationship groups**					
Continuous measure	314/6267	1.18 (1.11, 1.26)	<0.0001	1.14 (1.06, 1.23)	0.0004
Baseline NSS (all groups)					
Low NSS	172/3612	1.00		1.00	
High NSS	142/2727	1.73 (1.34, 2.24)	<0.0001	1.68 (1.22, 2.33)	0.0022
Change in measure (increase vs. no change)	75/1695	1.29 (0.93, 1.80)	0.1269	1.48 (1.03, 2.11)	0.0321
**Control of social life**					
Continuous measure	306/6172	0.86 (0.83, 0.88)	<0.0001	0.86 (0.83, 0.89)	<0.0001
Baseline control of social life					
Low control	162/1899	1.00		1.00	
High control	144/4342	0.43 (0.35, 0.54)	<0.0001	0.43 (0.34, 0.54)	<0.0001
Change in measure (decrease vs. no change)	44/1308	0.97 (0.68, 1.40)	0.8718	1.29 (0.86, 1.95)	0.2156
**Control of health**					
Continuous measure	306/6169	0.90 (0.85, 0.95)	0.0001	0.90 (0.85, 0.96)	0.0014
Baseline control of health					
Low control	172/2795	1.00		1.00	
High control	134/3444	0.69 (0.52, 0.90)	0.0074	0.71 (0.52, 0.96)	0.0286
Change in measure (decrease vs. no change)	46/1364	0.91 (0.63, 1.30)	0.5881	0.99 (0.67, 1.45)	0.9555
**Control of finances**					
Continuous measure	2309/6201	0.89 (0.85, 0.93)	<0.0001	0.90 (0.86, 0.94)	<0.0001
Baseline control of finances					
Low control	143/2207	1.00		1.00	
High control	166/4064	0.53 (0.41, 0.69)	<0.0001	0.55 (0.42, 0.73)	<0.0001
Change in measure (increase vs. no change)	59/1724	0.87 (0.65, 1.17)	0.348	1.06 (0.77, 1.47)	0.7022

Note: OR (95% CI): Odds ratios (95% confidence intervals); bold indicates *p*-value < 0.05; all models adjust for the complex sampling design of the HRS; ^c^ adjusted models control for race, toxic stress, and demographic factors; sex, education, alcohol consumption, smoking, BMI, moderate physical activity, retirement status and comorbidity due to diabetes, heart diseases, and stroke; ^d^ Cumulative stress is the sum of recent and life course events, sum can be 0–17. Measures of toxic stress and indicators of resilience were not mutually adjusted for one another in multivariable models.

## Data Availability

Publicly available datasets were analyzed in this study. These data can be found at https://www.hrs.isr,umich.edu/data-products/access-to-public-data, (accessed on 19 November 2021).

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
