# Peer review of "Toxic Psychosocial Stress, Resiliency Resources and Time to Dementia Diagnosis in a Nationally Representative Sample of Older Americans in the Health and Retirement Study from 2006–2016"

_ijerph, 2022, doi:10.3390/ijerph19042419_

Round 1
Reviewer 1 Report
This is a prospective, ten years follow-up, study carried out in a cohort (Health and Retirement Study) of 6516 “representative” American people ≥ 50 years old. The conclusion stated in the abstract “Policies that promote equitable treatment by race/ethnicity in social relations, health, justice, and economic systems may reduce TS and thus support cognitive reserve in older African Americans”, is a statement that we deeply feel to be right, but I’m not sure (unfortunately) that the data provide evidence for it. There are several potential limitations that I need to outline, as a well potential improvements to the manuscript.
- Included people are defined “Representative” Americans. I suppose the adjective needs to be supported by detailing criteria of inclusion.
- The outcome (dementia) is, defined as the answer to the questions “Has a doctor ever told you that you have a memory-related disease?” or “… Alzheimer’s disease or dementia”. This is quite a critical issue. The authors might well detail the outcome as well. For instance, is there any support to consider reliable the answer to the question?
- As for the methods to "measure" TS and RPF, the authors refer to previous publications. At least a summary of the methods should be reported.
As a general comment, there seems to be overconfidence in the results. For instance, the several reported 95% CI in the results section occasionally include the lower limit values very close to 1 (“discrimination-associated hazard of dementia diagnosis” CI: 1.01-3.03); “developing dementia amongst individuals that reported increased chronic stress relative to baseline” (CI: 1.02- 2.52); “risk of developing dementia among older Americans who reported an increase in NSS from all relationship groups relative to baseline” (CI: 1.03- 2.11).
The statistical approach deserves a further comment. Unfortunately, I do not have enough competence for a critical appraisal of the methodology. I just wonder about the power of the statistics when there are so many comparative evaluations. Table 2 report as many as 67 p values. Mind goes to the likelihood of having false positives. We often face the situation in which the profuse products of software packages go behind our capacity to provide a reasonable interpretation.
I also wonder about the confidence in correcting for so many sociodemographic confounders “… such as sex, marital status, retirement status, BMI, comorbidity, smoking status, and alcohol consumption”.
Reviewer 2 Report
The premise of this research and the statistical approach are both sound. The cohort size is large, although very White (83%). It is remarkable that only 5% had a dementia dx, given that rates of dementia in 2014 were 13.8% for African Americans and 10% for Whites in the general population(CDC). That 27% of the study population had 3+ comorbid conditions is consistent with 2018 data presented by Boersma, et. al. It would be helpful to include national data for comparisons in the Introduction.
Your choice of words in the Discussion section may serve to inflate how your findings are considered. Earlier in your manuscript your describe toxic stress and how it was dichotomized into no stress or some measure of stress (1+), yet you describe it as High Toxic Stress in the Discussion. I suggest removing the word "high" throughout the discussion and use toxic stress instead. In Lines 301-303, you state, "These findings are in line with our study hypothesis that TS in aging adults will accelerate cognitive decline, as manifested by faster onset of dementia." Your hypothesis as stated in Lines 73-74 was, : "(a) higher levels of TS and lower levels of RPF will be associated
with earlier dementia diagnosis in older American adults". It would be more accurate to state in the Discussion that findings showed that higher levels of TS were associated with an earlier dementia diagnosis. Finally, more should be made of the other factors associated with incident dementia on Lines 278-281. These are all lifestyle-related factors that may be related to race and education and contribute to the burden of disease related to dementia.
Round 2
Reviewer 1 Report
The manuscript appreciably includes parts of the missing information on the methods and some claims are less assertive. There are not substantial changes, however, and the statistical issue has been dealt with just with verbal reassurance.
